# Interpretable Counting for Visual Question Answering

**Alexander Trott, Caiming Xiong,**[*] **& Richard Socher**
Salesforce Research
Palo Alto, CA
`{atrott,cxiong,rsocher}@salesforce.com`

## Abstract

Questions that require counting a variety of objects in images remain a major challenge in visual question answering (VQA). The most common approaches to VQA involve either classifying answers based on fixed length representations of both the image and question or summing fractional counts estimated from each section of the image. In contrast, we treat counting as a sequential decision process and force our model to make discrete choices of what to count. Specifically, the model sequentially selects from detected objects and learns interactions between objects that influence subsequent selections. A distinction of our approach is its intuitive and interpretable output, as discrete counts are automatically grounded in the image. Furthermore, our method outperforms the state of the art architecture for VQA on multiple metrics that evaluate counting.

## 1 Introduction

Visual question answering (VQA) is an important benchmark to test for context-specific reasoning over complex images. While the field has seen substantial progress, counting-based questions have seen the least improvement (Chattopadhyay et al., 2017). Intuitively, counting should involve finding the number of distinct scene elements or objects that meet some criteria, see Fig. 1 for an example. In contrast, the predominant approach to VQA involves representing the visual input with the final feature map of a convolutional neural network (CNN), attending to regions based on an encoding of the question, and classifying the answer from the attention-weighted image features (Xu & Saenko, 2015; Yang et al., 2015; Xiong et al., 2016; Lu et al., 2016b; Fukui et al., 2016; Kim et al., 2017). Our intuition about counting seems at odds with the effects of attention, where a weighted average obscures any notion of distinct elements. As such, we are motivated to re-think the typical approach to counting in VQA and propose a method that embraces the discrete nature of the task.

Our approach is partly inspired by recent work that represents images as a set of distinct objects, as identified by object detection (Anderson et al., 2017), and making use of the relationships between these objects (Teney et al., 2016). We experiment with counting systems that build off of the vision module used for these two works, which represents each image as a set of detected objects. For training and evaluation, we create a new dataset, HowMany-QA. It is taken from the counting-specific union of VQA 2.0 (Goyal et al., 2017) and Visual Genome QA (Krishna et al., 2016).

We introduce the Interpretable Reinforcement Learning Counter (IRLC), which treats counting as a sequential decision process. We treat learning to count as learning to enumerate the relevant objects in the scene. As a result, IRLC not only returns a count but also the objects supporting its answer. This output is produced through an iterative method. Each step of this sequence has two stages: First, an object is selected to be added to the count. Second, the model adjusts the priority given to unselected objects based on their configuration with the selected objects (Fig. 1). We supervise only the final count and train the decision process using reinforcement learning (RL).

Additional experiments highlight the importance of the iterative approach when using this manner of weak supervision. Furthermore, we train the current state of the art model for VQA on HowMany-QA and find that IRLC achieves a higher accuracy and lower count error. Lastly, we compare the

---

[*]Corresponding author

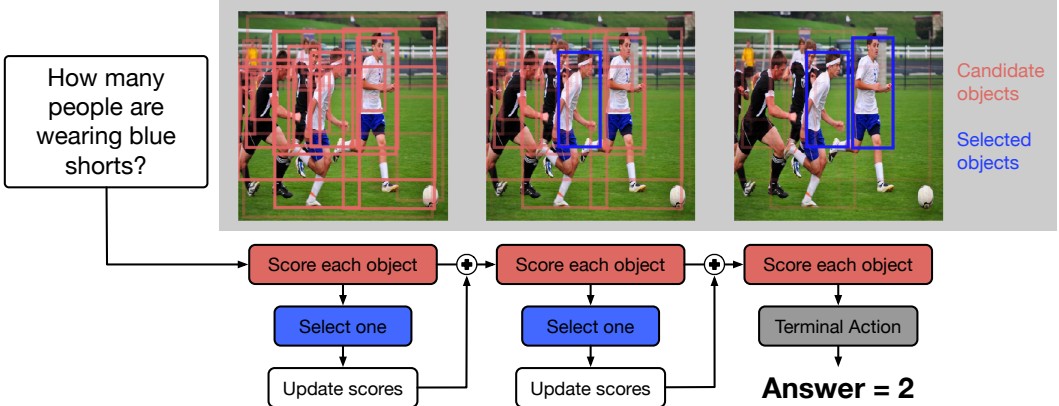

Figure 1: IRLC takes as input a counting question and image. Detected objects are added to the returned count through a sequential decision process. The above example illustrates actual model behavior after training.

grounded counts of our model to the attentional focus of the state of the art baseline to demonstrate the interpretability gained through our approach.

## 2 RELATED WORK

**Visual representations for counting.** As a standalone problem, counting from images has received some attention but typically within specific problem domains. Segui et al. (2015) explore training a CNN to count directly from synthetic data. Counts can also be estimated by learning to produce density maps for some category of interest (typically people), as in Lempitsky & Zisserman (2010); Oñoro-Rubio & López-Sastre (2016); Zhang et al. (2015). Density estimation simplifies the more challenging approach of counting by instance-by-instance detection (Ren & Zemel, 2017). Methods to detect objects and their bounding boxes have advanced considerably (Girshick et al., 2015; Girshick, 2015; Ren et al., 2015b; Dai et al., 2016; Lin et al., 2017) but tuning redundancy reduction steps in order to count is unreliable (Chattopadhyay et al., 2017). Here, we overcome this limitation by allowing flexible, question specific interactions during counting.

Alternative approaches attempt to model subitizing, which describes the human ability to quickly and accurately gauge numerosity when at most a few objects are present. Zhang et al. (2017) demonstrates that CNNs may be trained towards a similar ability when estimating the number of salient objects in a scene. This approach was extended to counting 80 classes of objects simultaneously in Chattopadhyay et al. (2017). Their model is trained to estimate counts within each subdivision of the full image, where local counts are typically within the subitizing range.

The above studies apply counting to a fixed set of object categories. In contrast, we are interested in counting during visual question answering, where the criteria for counting change from question to question and can be arbitrarily complex. This places our work in a different setting than those that count from the image alone. For example, Chattopadhyay et al. (2017) apply their trained models to a subset of VQA questions, but their analysis was limited to the specific subset of examples where the question and answer labels agreed with the object detection labels they use for training. Here, we overcome this limitation by learning to count directly from question/answer pairs.

**Visual question answering.** The potential of deep learning to fuse visual and linguistic reasoning has been recognized for some time (Socher et al., 2014; Lu et al., 2016a). Visual question answering poses the challenge of retrieving question-specific information from an associated image, often requiring complex scene understanding and flexible reasoning. In recent years, a number of datasets have been introduced for studying this problem (Malinowski & Fritz, 2014; Ren et al., 2015a; Zhu et al., 2015; Agrawal et al., 2015; Goyal et al., 2017; Krishna et al., 2016). The majority of recent progress has been aimed at the so-named "VQA" datasets (Agrawal et al., 2015; Goyal et al., 2017),

where counting questions represent roughly 11% of the data. Though our focus is on counting questions specifically, prior work on VQA is highly relevant.

An early baseline for VQA represents the question and image at a coarse granularity, respectively using a "bag of words" embedding along with spatially-pooled CNN outputs to classify the answer (Zhou et al., 2015). In Ren et al. (2015a), a similar fixed-length image representation is fused with the question embeddings as input to a recurrent neural network (RNN), from which the answer is classified.

**Attention.** More recent variants have chosen to represent the image at a finer granularity by omitting the spatial pooling of the CNN feature map and instead use attention to focus relevant image regions before producing an answer (Xu & Saenko, 2015; Yang et al., 2015; Xiong et al., 2016; Lu et al., 2016b; Fukui et al., 2016; Kim et al., 2017). These works use the spatially-tiled feature vectors output by a CNN to represent the image; others follow the intuition that a more meaningful representation may come from parsing the feature map according to the locations of objects in the scene (Shih et al., 2015; Ilievski et al., 2016). Notably, using object detection was a key design choice for the winning submission for the VQA 2017 challenge (Anderson et al., 2017; Teney et al., 2017). Work directed at VQA with synthetic images (which sidesteps the challenges created by computer vision) has further demonstrated the utility that relationships may provide as an additional form of image annotation (Teney et al., 2016).

**Interpretable VQA.** The use of "scene graphs" in real-image VQA would have the desirable property that intermediate model variables would be grounded in concepts explicitly, a step towards making neural reasoning more transparent. A conceptual parallel to this is found in Neural Module Networks (Andreas et al., 2016a;b; Hu et al., 2017), which gain interpretability by grounding the reasoning process itself in defined concepts. The general concept of interpretable VQA has been the subject of recent interest. Park et al. (2016) extends the task itself to include generating explanations for produced answers. Chandrasekaran et al. (2017) take a different approach, asking how well humans can learn patterns in answers and failures of a trained VQA model. While humans indeed identify some patterns, they do not gain any apparent insight from knowing intermediate states of the model (such as its attentional focus). In light of this, we are motivated by the goal of developing more transparent AI.

We address this at the level of counting in VQA. We show that, despite the challenge presented by this particular task, an intuitive approach gains in both performance and interpretability over state of the art.

## 3 DATASETS

Within the field of VQA, the majority of progress has been aimed at the VQA dataset (Agrawal et al., 2015) and, more recently, VQA 2.0 (Goyal et al., 2017), which expands the total number of questions in the dataset and attempts to reduce bias by balancing answers to repeated questions. VQA 2.0 consists of 1.1M questions pertaining to the 205K images from COCO (Lin et al., 2014). The examples are divided according to the official COCO splits.

In addition to VQA 2.0, we incorporate the Visual Genome (VG) dataset (Krishna et al., 2016). Visual Genome consists of 108K images, roughly half of which are part of COCO. VG includes its own visual question answering dataset. We include examples from that dataset when they pertain to an image in the VQA 2.0 training set.

### 3.1 HOWMANY-QA

In order to evaluate counting specifically, we define a subset of the QA pairs, which we refer to as HowMany-QA. Our inclusion criteria were designed to filter QA pairs where the question asks for a count, as opposed to simply an answer in the form of a number (Fig 2). For the first condition, we require that the question contains one of the following phrases: "how many", "number of", "amount of", or "count of". We also reject a question if it contains the phrase "number of the", since this phrase frequently refers to a printed number rather than a count (i.e. "what is the number of the bus?"). Lastly, we require that the ground-truth answer is a number between 0 to 20 (inclusive). The original VQA 2.0 train set includes roughly 444K QA pairs, of which 57,606 are labeled as having

What time is on the
clock?
**ground truth = 9:35**
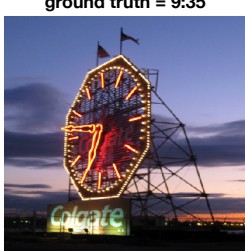

What is the age of the
man?
**ground truth = 60**
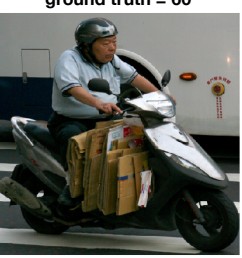

How many people does
the jet seat?
**ground truth = 200**
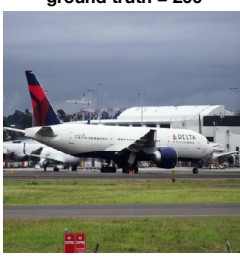

What number is the
batter?
**ground truth = 59**
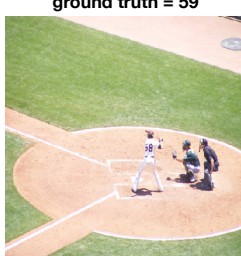

Figure 2: Examples of question-answer pairs that are excluded from HowMany-QA. This selection exemplifies the common types of "number" questions that do not require counting and therefore distract from our objective: (from left to right) time, general number-based answers, ballparking, and reading numbers from images. Importantly, the standard VQA evaluation metrics do not distinguish these from counting questions; instead, performance is reported for "number" questions as a whole.

a "number" answer. Focusing on counting questions results in a still very large dataset with 47,542 pairs, showing the importance of this subtask.

Due to our filter and focus on counting questions, we cannot make use of the official test data since its annotations are not available. Hence, we divide the validation data into separate development and test sets. More specifically, we apply the above criteria to the official validation data and select 5,000 of the resulting QA pairs to serve as the test data. The remaining 17,714 QA pairs are used as the development set.

| Split | QA Pairs | Images |
|---|---|---|
| Train | 83,642 | 31,932 |
| *from VQA 2.0* | 47,542 | 31,932 |
| *from VG* | 36,100 | 0 |
| Dev. | 17,714 | 13,119 |
| Test | 5,000 | 2,483 |

Table 1: Size breakdown of HowMany-QA. Neither development or test included VG data.

As mentioned above, the HowMany-QA training data is augmented with available QA pairs from Visual Genome, which are selected using the same criteria. A breakdown of the size and composition of HowMany-QA is provided in Table 1. All models compared in this work are trained and evaluated on HowMany-QA. To facilitate future comparison to our work, we have made the training, development, and test question IDs available for download.

## 4 MODEL

In this work, we focus specifically on counting in the setting of visual question answering (where the criteria for counting changes on a question-by-question basis). In addition, we are interested in model interpretability. We explore this notion by experimenting with models that are capable of producing question-guided counts which are visually grounded in object proposals.

Rather than substantially modifying existing counting approaches – such as Chattopadhyay et al. (2017) – we compare three models whose architectures naturally fit within our experimental scope. These models each produce a count from the outputs of an object detection module and use identical strategies to encode the question and compare it to the detected objects. The models differ only in terms of how these components are used to produce a count (Fig. 3a).

### 4.1 OBJECT DETECTION

Our approach is inspired by the strategy of Anderson et al. (2017) and Teney et al. (2017). Their model, which represents current state of the art in VQA, infers objects as the input to the question-answering system. This inference is performed using the Faster R-CNN architecture (Ren et al., 2015b). The Faster R-CNN proposes a set of regions corresponding to objects in the image. It

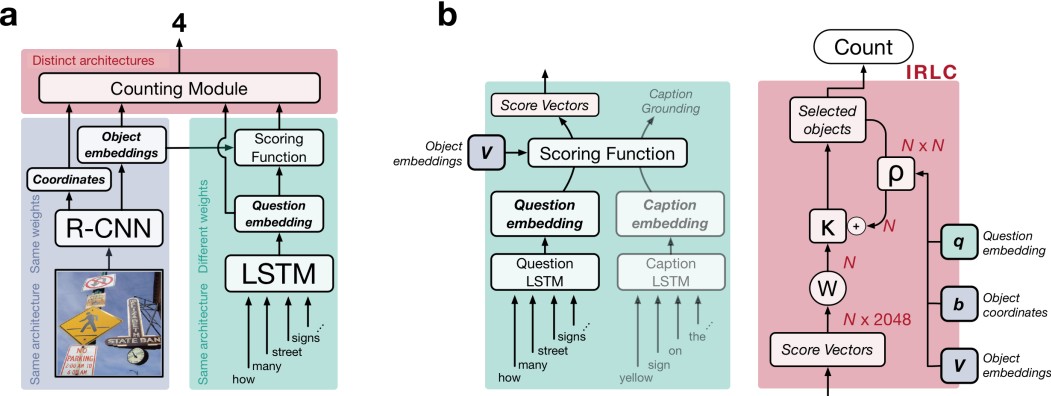

Figure 3: **(a)** Each model includes three basic modules: vision (blue), language (green), and counting (red). Text in the shaded regions describes which aspects of these modules are shared across models. **(b)** (left) The language model embeds the question and compares it to each object using a scoring function, which is jointly trained with caption grounding; (right) IRLC counting module.

encodes the image as a set of bounding boxes $\{b_1, ..., b_N\}$, $b_i \in \mathbb{R}^4$ and complementary set of object encodings $\{v_1, ..., v_N\}$, $v_i \in \mathbb{R}^{2048}$, corresponding to the locations and feature representations of each of the $N$ detected objects, respectively (blue box in Fig. 3a).

Rather than train our own vision module from scratch, we make use of the publicly available object proposals learned in Anderson et al. (2017). These provide rich, object-centric representations for each image in our dataset. These representations are fixed when learning to count and are shared across each of the QA models we experiment with.

## 4.2 LANGUAGE

Each architecture encodes the question and compares it to each detected object via a scoring function. We define $q$ as the final hidden state of an LSTM (Hochreiter & Schmidhuber, 1997) after processing the question and compute a score vector for each object (Fig. 3, green boxes):

$$h^t = \text{LSTM}\left(x^t, h^{t-1}\right) \qquad q = h^T \tag{1}$$

$$s_i = f^S\left([q, v_i]\right) \tag{2}$$

Here, $x_t$ denotes the word embedding of the question token at position $t$ and $s_i \in \mathbb{R}^n$ denotes the score vector encoding the relevance of object $i$ to the question. Following Anderson et al. (2017), we implement the scoring function $f^S : \mathbb{R}^m \to \mathbb{R}^n$ as a layer of Gated Tanh Units (GTU) (van den Oord et al., 2016). $[,]$ denotes vector concatenation.

We experiment with jointly training the scoring function to perform caption grounding, which we supervise using region captions from Visual Genome. Region captions provide linguistic descriptions of localized regions within the image, and the goal of caption grounding is to identify which object a given caption describes (details provided in Section B.1 of the Appendix). Caption grounding uses a strategy identical to that for question answering (Eqs. 1 and 2): an LSTM is used to encode each caption and the scoring function $f^S$ is used to encode its relevance to each detected object. The weights of the scoring function are tied for counting and caption grounding (Fig. 3b). We include results from experiments where caption grounding is ignored.

## 4.3 COUNTING

**Interpretable RL Counter (IRLC).** For our proposed model, we aim to learn *how* to count by learning *what* to count. We assume that each counting question implicitly refers to a subset of the objects within a scene that meet some variable criteria. In this sense, the goal of our model is to enumerate that subset of objects.

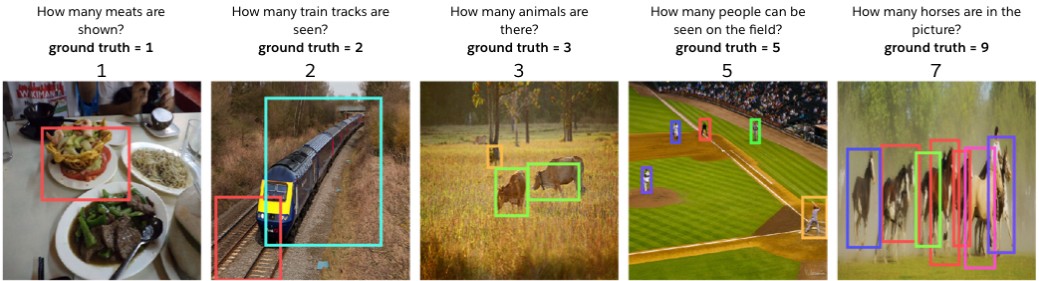

Figure 4: Grounded counts produced by IRLC. Counts are formed from selections of detected objects. Each image displays the objects that IRLC chose to count.

To implement this as a sequential decision process, we need to represent the probability of selecting a given action and how each action affects subsequent choices. To that end, we project the object scores $s \in \mathbb{R}^{N \times n}$ to a vector of logits $\kappa \in \mathbb{R}^N$, representing how likely each object is to be counted, where $N$ is the number of detected objects:

$$\kappa = Ws + b \tag{3}$$

And we compute a matrix of interaction terms $\rho \in \mathbb{R}^{N \times N}$ that are used to update the logits $\kappa$. The value $\rho_{ij}$ represents how selecting object $i$ will change $\kappa_j$. We calculate this interaction from a compressed representation of the question ($Wq$), the dot product of the normalized object vectors ($\hat{v}_i^{\mathrm{T}} \hat{v}_j$), the object coordinates ($b_i$ and $b_j$), and basic overlap statistics ($\mathrm{IoU}_{ij}, \mathrm{O}_{ij}$, and $\mathrm{O}_{ji}$):

$$\rho_{ij} = f^\rho \left( \left[ Wq, \hat{v}_i^{\mathrm{T}} \hat{v}_j, b_i, b_j, \mathrm{IoU}_{ij}, \mathrm{O}_{ij}, \mathrm{O}_{ji} \right] \right) \tag{4}$$

where $f^\rho : x \in \mathbb{R}^m \Rightarrow \mathbb{R}$ is a 2-layer MLP with ReLU activations.

For each step $t$ of the counting sequence we greedily select the action with the highest value (interpreted as either selecting the next object to count or terminating), and update $\kappa$ accordingly:

$$a^t = \mathrm{argmax}_i \left[ \kappa^t, \zeta \right] \tag{5}$$

$$\kappa^{t+1} = \kappa^t + \rho(a^t, \cdot) \tag{6}$$

where $\zeta$ is a learnable scalar representing the logit value of the terminal action, and $\kappa^0$ is the result of Equation 3 (Fig. 3b, red box). The action $a^t$ is expressed as the index of the selected object. $\rho(a^t, \cdot)$ denotes the row of $\rho$ indexed by $a^t$. Each object is only allowed to be counted once. We define the count $C$ as the timestep when the terminal action was selected $t : a^t = N + 1$.

This approach bears some similarity to Non-Maximal Suppression (NMS), a staple technique in object detection to suppress redundant proposals. However, our approach is far less rigid and allows the question to determine how similar and/or overlapping objects interact.

**Training IRLC.** Because the process of generating a count requires making discrete decisions, training requires that we use techniques from Reinforcement Learning. Given our formulation, a natural choice is to apply REINFORCE (Williams, 1992). To do so, we calculate a distribution over action probabilities $p^t$ from $\kappa^t$ and generate a count by iteratively sampling actions from the distribution:

$$p^t = \mathrm{softmax} \left( \left[ \kappa^t, \zeta \right] \right) \qquad a^t \sim p^t \tag{7}$$

$$\kappa^{t+1} = \kappa^t + \rho(a^t, \cdot) \tag{8}$$

We calculate the reward using Self-Critical Sequence training (Rennie et al., 2017; Anderson et al., 2017; Paulus et al., 2017), a variation of policy gradient. We define $E = |C - C^{\mathrm{GT}}|$ to be the count error and define the reward as $R = E^{\mathrm{greedy}} - E$, where $E^{\mathrm{greedy}}$ is the baseline count error obtained by greedy action selection (which is also how the count is measured at test time). From this, we define our (unnormalized) counting loss as

$$\tilde{L}_C = -R \sum_t \log p^t \left( a^t \right) \tag{9}$$

Additionally, we include two auxiliary objectives to aid learning. For each sampled sequence, we measure the total negative policy entropy $H$ across the observed time steps. We also measure the average interaction strength at each time step and collect the total

$$\tilde{P}_{\mathrm{H}} = -\sum_t H\left(p^t\right) \qquad \tilde{P}_{\mathrm{I}} = \sum_{i \in \{a^0 \dots a^t\}} \frac{1}{N} \sum_j L_1\left(\rho_{ij}\right) \tag{10}$$

where $L_1$ is the Huber loss from Eq 12. Including the entropy objective is a common strategy when using policy gradient (Williams & Peng, 1991; Minh et al., 2016; Luo et al., 2017) and is used to improve exploration. The interaction penalty is motivated by the *a priori* expectation that interactions should be sparse. During training, we minimize a weighted sum of the three losses, normalized by the number of decision steps. As before, we provide training and implementation details in the Appendix (Sec. B.2).

**SoftCount.** As a baseline approach, we train a model to count directly from the outputs $s$ of the scoring function. For each object, we project its score vector $s_i$ to a scalar value and apply a sigmoid nonlinearity, denoted as $\sigma$, to assign the object a count value between 0 and 1. The total count is the sum of these fractional, object-specific count values. We train this model by minimizing the Huber loss associated with the absolute difference $e$ between the predicted count $C$ and the ground truth count $C^{\mathrm{GT}}$:

$$C = \sum_i \sigma\left(W s_i\right) \tag{11}$$

$$L_1 = \begin{cases} 0.5 e^2 & \text{if } e \leq 1 \\ e - 0.5 & \text{otherwise} \end{cases} \qquad e = |C - C^{\mathrm{GT}}| \tag{12}$$

For evaluation, we round the estimated count $C$ to the nearest integer and limit the output to the maximum ground truth count (in this case, 20).

**Attention Baseline (UpDown).** As a second baseline, we re-implement the QA architecture introduced in Anderson et al. (2017), which the authors refer to as UpDown – see also Teney et al. (2017) for additional details. We focus on this architecture for three main reasons. First, it represents the current state of the art for VQA 2.0. Second, it was designed to use the visual representations we employ. And, third, it exemplifies the common two-stage approach of (1) deploying question-based attention over image regions (here, detected objects) to get a fixed-length visual representation

$$\alpha = \mathrm{softmax}\left(W s\right); \quad \hat{v} = \sum \alpha_i v_i \tag{13}$$

and then (2) classifying the answer based on this average and the question encoding

$$v' = f^V\left(\hat{v}\right); \quad q' = f^Q\left(q\right) \tag{14}$$

$$p = \mathrm{softmax}\left(f^C\left(v' \otimes q'\right)\right) \tag{15}$$

where $s \in \mathbb{R}^{N \times n}$ denotes the matrix of score vectors for each of the $N$ detected objects and $\alpha \in \mathbb{R}^N$ denotes the attention weights. Here, each function $f$ is implemented as a GTU layer and $\otimes$ denotes element-wise multiplication. For training, we use a cross entropy loss, with the target given by the ground-truth count. At test time, we use the most probable count given by $p$.

## 5 RESULTS

### 5.1 COUNTING PERFORMANCE

We use two metrics for evaluation. For consistency with past work, we report the standard VQA test metric of accuracy. Since accuracy does not measure the degree of error we also report root-mean-squared-error (RMSE), which captures the typical deviation between the estimated and ground-truth count and emphasizes extreme errors. Details are provided in the Appendix (Sec. D).

To better understand the performance of the above models, we also report the performance of two non-visual baselines. The first baseline (Guess1) shows the performance when the estimated count is always 1 (the most common answer in the training set). The second baseline (LSTM) learns to predict the count directly from a linear projection of the question embedding $q$ (Eq. 1).

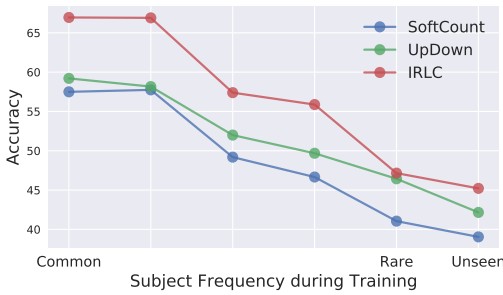 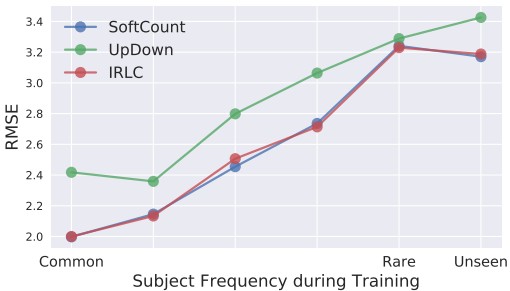

Figure 5: Model performance on the HowMany-QA development set, grouped according to the frequency with which the counting subject appeared in the training data.

IRLC achieves the highest overall accuracy and (with SoftCount) the lowest overall RMSE on the test set (Table 2). Interestingly, SoftCount clearly lags in accuracy but is competitive in RMSE, arguing that accuracy and RMSE are not redundant. We observe this to result from the fact that IRLC is less prone to small errors and very slightly more prone to large errors (which disproportionately impact RMSE). However, whereas UpDown improves in accuracy at the cost of RMSE, IRLC is substantially more accurate without sacrificing overall RMSE.

| Model | Accuracy | RMSE |
|---|---|---|
| Guess1 | 33.8 | 3.74 |
| LSTM | 36.8 | 3.47 |
| SoftCount | 50.2 (49.2) | **2.37** (2.45) |
| UpDown | 52.7 (51.5) | 2.64 (2.69) |
| IRLC | **57.7** (56.1) | **2.37** (2.45) |

Table 2: HowMany-QA test set performance. Values in parentheses apply to models trained without caption grounding.

To gain more insight into the performance of these models, we calculate these metrics within the development set after separating the data according to how common the subject of the count is during training[1]. We break up the questions into 5 roughly equal-sized bins representing increasingly uncommon subjects. We include a 6th bin for subjects never seen during training. The accuracy and RMSE across the development set are reported for each of these bins in Figure 5.

Organizing the data this way reveals two main trends. First, all models perform better when asked to count subjects that were common during training. Second, the performance improvements offered by IRLC over UpDown persist over all groupings of the development data.

## 5.2 GROUNDING QUALITY

We introduce a novel analysis to quantify how well counted objects match the subject of the question. To perform this analysis, we form generic questions that refer to the object categories in the COCO object detection dataset. We take the object proposals counted in response to a given question and compare them to the ground truth COCO labels to determine how relevant the counted object proposals are. Our metric takes on a value of 1 when the counted objects perfectly map onto the category to which the question refers. Values around 0 indicate that the counted objects were not relevant to the question. Section D of the Appendix details how the grounding quality metric is calculated.

We perform this analysis for each of the 80 COCO categories using the images in the HowMany-QA development set. Figure 6 compares the grounding quality of SoftCount and IRLC, where each point represents the average grounding quality for a particular COCO category. As with the previous two metrics, grounding quality is highest for COCO categories that are more common during training. We observe that IRLC consistently grounds its counts in objects that are more relevant to the question than does SoftCount. A paired t-test shows that this trend is statistically significant ($p < 10^{-15}$).

---

[1]We infer the subject using a simple dependency-parsing heuristic. Specifically, we extract the root word of the first noun-chunk.

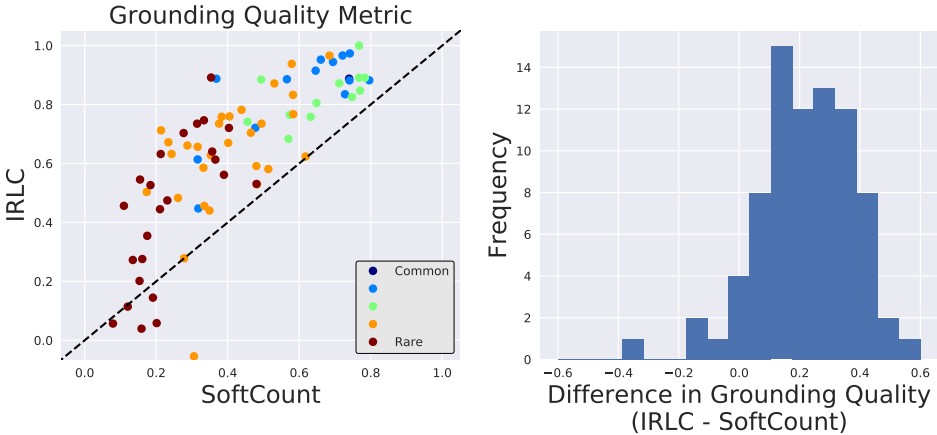

Figure 6: (Left) Average grounding quality for each of the COCO object categories, as measured for SoftCount and IRLC. Each point represents a COCO category and is colored according to how common the category was during training (as in Figure 5). (Right) Histogram showing the difference in grounding quality between IRLC and SoftCount.

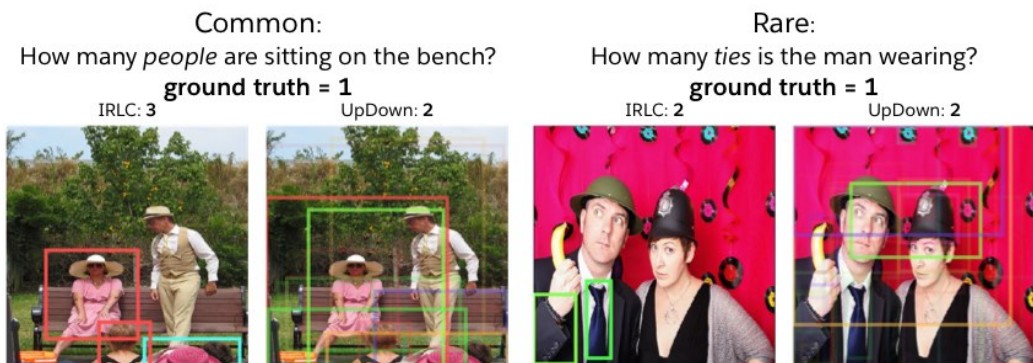

Figure 7: Examples of failure cases with common and rare subjects ("people" and "ties," respectively). Each example shows the output of IRLC, where boxes correspond to counted objects, and the output of UpDown, where boxes are shaded according to their attention weights.

## 5.3 QUALITATIVE ANALYSIS

The design of IRLC is inspired by the ideal of interpretable VQA (Chandrasekaran et al., 2017). One hallmark of interpretability is the ability to predict failure modes. We argue that this is made more approachable by requiring IRLC to identify the objects in the scene that it chooses to count.

Figure 7 illustrates two failure cases that exemplify observed trends in IRLC. In particular, IRLC has little trouble counting people (they are the most common subject) but encounters difficulty with referring phrases (in this case, "sitting on the bench"). When asked to count ties (a rare subject), IRLC includes a sleeve in the output, demonstrating the tendency to misidentify objects with few training examples. These failures are obvious by virtue of the grounded counts, which point out exactly which objects IRLC counted. In comparison, the attention focus of UpDown (representing the closest analogy to a grounded output) does not identify any pattern. From the attention weights, it is unclear which scene elements form the basis of the returned count.

Indeed, the two models may share similar deficits. We observe that, in many cases, they produce similar counts. However, we stress that without IRLC and the chance to observe such similarities such deficits of the UpDown model would be difficult to identify.

The Appendix includes further visualizations and comparisons of model output, including examples of how IRLC uses the iterative decision process to produce discrete, grounded counts (Sec. A).

# 6 CONCLUSION

We present an interpretable approach to counting in visual question answering, based on learning to enumerate objects in a scene. By using RL, we are able to train our model to make binary decisions about whether a detected object contributes to the final count. We experiment with two additional baselines and control for variations due to visual representations and for the mechanism of visual-linguistic comparison. Our approach achieves state of the art for each of the evaluation metrics. In addition, our model identifies the objects that contribute to each count. These groundings provide traction for identifying the aspects of the task that the model has failed to learn and thereby improve not only performance but also interpretability.

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

## A EXAMPLES

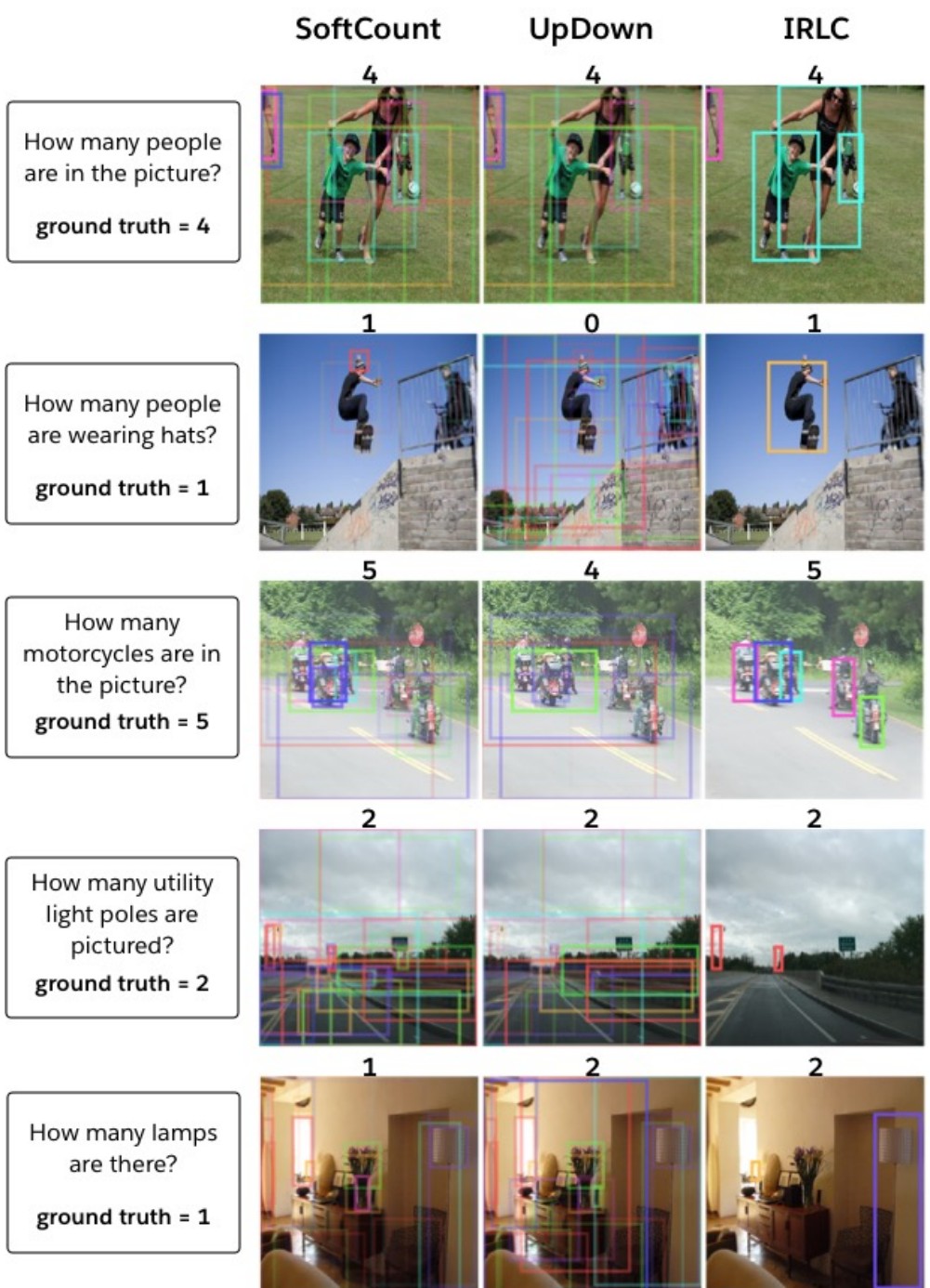

Figure 8: Example outputs produced by each model. For SoftCount, objects are shaded according to the fractional count of each (0=transparent; 1=opaque). For UpDown, we similarly shade the objects but use the attention focus to determine opacity. For IRLC, we plot only the boxes from objects that were selected as part of the count.

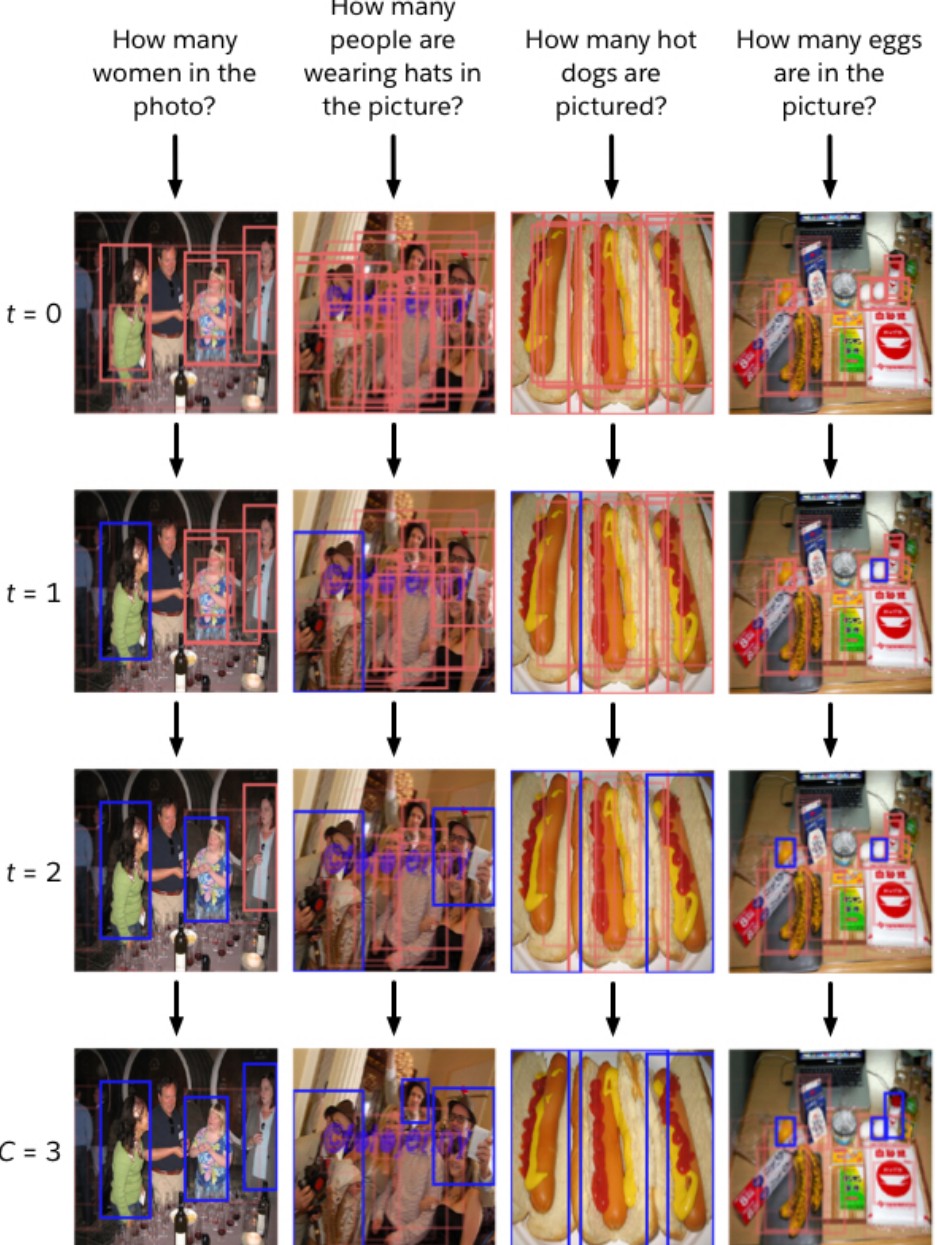

Figure 9: Sequential counting of IRLC. At each timestep, we illustrate the unchosen boxes in pink, and shade each box according to $\kappa^t$ (corresponding to the probability that the box would be selected at that time step; see main text). We also show the already-selected boxes in blue. For each of the questions, the counting sequence terminates at $t = 3$, meaning that the returned count $C$ is 3. For each of these questions, that is the correct answer. The example on the far right is a 'correct failure,' a case where the correct answer is returned but the counted objects are not related to the question. These kinds of subtle failures are revealed with the grounded counts.

## B   Training and implementation details

### B.1   Caption Grounding

We experiment with jointly training counting and caption grounding. The goal of caption grounding is, given a set of objects and a caption, to identify the object that the caption describes. Identical to the first stages of answering the counting question, we use an LSTM to encode the caption and compare it to each of the objects using the scoring function:

$$h^t = \text{LSTM}\left(x^t, h^{t-1}\right) \tag{16}$$

$$s_i = f^S\left(\left[h^T, v_i\right]\right) \tag{17}$$

where $h \in \mathbb{R}^{1024}$, $x_i^t \in \mathbb{R}^{300}$ is the embedding for the token at timestep $t$ of the caption, $T$ is the caption length, and $f^S : \mathbb{R}^m \to \mathbb{R}^n$ is the scoring function (Sec. 4.3). The embedding of object $i$ is denoted by $v_i$ and the relevance of this object to the caption is encoded by the score vector $s_i$.

We project each such score vector to a scalar logit $\alpha_i$ and apply a softmax nonlinearity to estimate $p \in \mathbb{R}^N$, where $N$ is the number of object proposals and $p_i$ denotes the probability that the caption describes object proposal $i$:

$$\alpha_i = W s_i + b \tag{18}$$

$$p = \text{softmax}\left(\alpha\right). \tag{19}$$

During training, we randomly select four of the images in the batch of examples to use for caption grounding (rather than the full 32 images that make up a batch). To create training data from the region captions in Visual Genome, we assign each caption to one of the detected object proposals. To do so, we compute the intersection over union between the ground truth region that the caption describes and the coordinates of each object proposal. We assign the object proposal with the largest IoU to the caption. If the maximum IoU for the given caption is less than 0.5, we ignore it during training. We compute the grounding probability $p$ for each caption to which we can successfully assign a detection and train using the cross entropy loss averaged over the captions. We weight the loss associated with caption grounding by 0.1 relative to the counting loss.

### B.2   Counting models

Each of the considered counting models makes use of the same basic architecture for encoding the question and comparing it with each of the detected objects. For each model, we initialized the word embeddings from GloVe (Pennington et al., 2014) and encoded the question with an LSTM of hidden size 1024. The only differences in the model-specific implementations of the language module was the hidden size of the scoring function $f^S$. We determined these specifics from the optimal settings observed during initial experiments. We use a hidden size of 512 for SoftCount and UpDown and a hidden size of 2048 for IRLC. We observed that the former two models were more prone to overfitting, whereas IRLC benefited from the increased capacity.

When training on counting, we optimize using Adam (Kingma & Ba, 2014). For SoftCount and UpDown, we use a learning rate of $3\text{x}10^{-4}$ and decay the learning rate by 0.8 when the training accuracy plateaus. For IRLC, we use a learning rate of $5\text{x}10^{-4}$ and decay the learning rate by 0.99999 every iteration. For all models, we regularize using dropout and apply early stopping based on the development set accuracy (see below).

When training IRLC, we apply the sampling procedure 5 times per question and average the losses. We weight the entropy penalty $P_H$ and interaction penalty $P_I$ (Eq. 10) both by 0.005 relative to the counting loss. These penalty weights yield the best development set accuracy within the hyperparameter search we performed (Fig. 10).

## C   Additional analyses and experiments

**IRLC auxiliary loss.** We performed a grid search to determine the optimal setting for the weights of the auxiliary losses for training IRLC. From our observations, the entropy penalty is important to balance the exploration of the model during training. In addition, the interaction penalty prevents

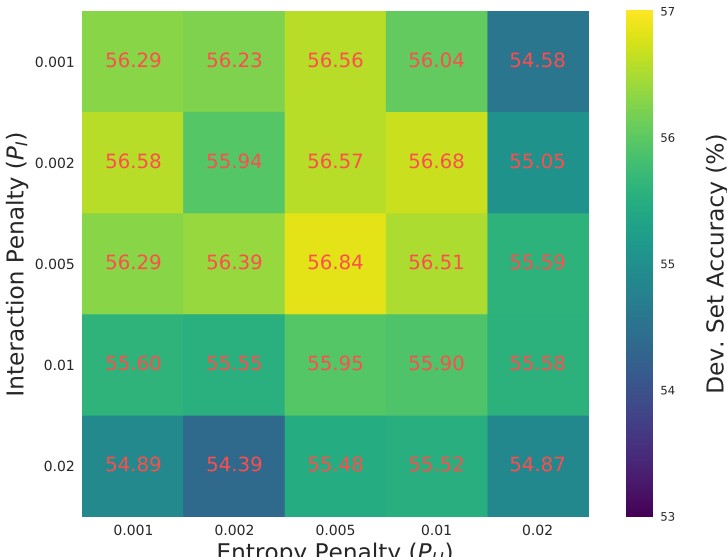

Figure 10: Results of a hyperparameter sweep over the penalty weights. The accuracy over the development set is reported for each weight setting.

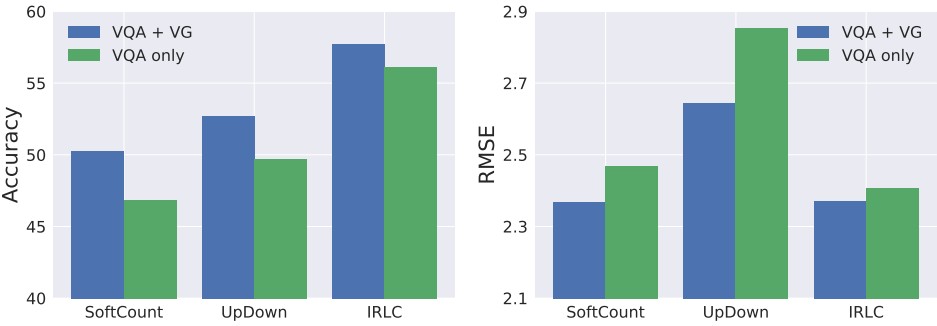

Figure 11: HowMany-QA test set performance for models trained with the full HowMany-QA training data (blue) and trained without the additional data from Visual Genome (green).

degenerate counting strategies. The results of the grid search suggest that these auxiliary losses improve performance but become unhelpful if given too much weight (Fig. 10). In any case, IRLC outperforms the baseline models across the range of settings explored.

**Data augmentation with Visual Genome.** Here, we compare performance on the HowMany-QA test set for models trained with and without additional data from Visual Genome QA. In all cases, performance benefits from the additional training data (Fig. 11). On average, excluding Visual Genome from the training data decreases accuracy by 2.7% and increases RMSE by 0.12. Interestingly, the performance of IRLC is most robust to the loss of training data.

**Ordinality of UpDown output.** Whereas the training objectives for SoftCount and IRLC intrinsically reflect the ordinal nature of counts, the same is not true for UpDown. For example, the loss experienced by SoftCount and IRLC reflect the degree of error between the estimated count and the ground truth target; however, UpDown is trained only to place high probability mass on the ground truth value (missing by 1 or by 10 are treated as equally incorrect). We examine the patterns in the output count probabilities from UpDown to ask whether the model learns an ordinal representation despite its non-ordinal training objective. Figure 12 illustrates these trends. When the estimated count is less than 5, the second-most probable count is very frequently adjacent to the most probable count. When the estimated count is larger than 5, the probability distribution is less smooth, such

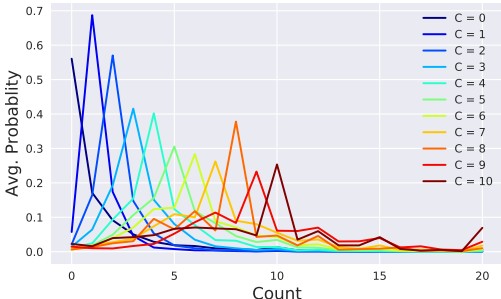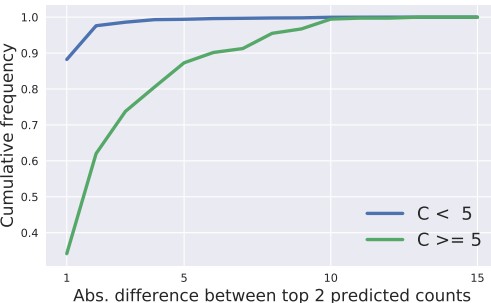

Figure 12: (Left) Average count probability (Eq. 15) from UpDown, grouped according to the estimated count. (Right) Cumulative distribution of the absolute difference between the top two predicted counts, shown for when the most likely count was less than 5 (blue) and when it was greater than or equal to 5 (green). The probability distributions are much less smooth when the estimated count is large.

that the second-most probable count is often considerably different than the most probable count. This result suggests that UpDown learns ordinality for lower count values (where training data is abundant) but fails to generalize this concept to larger counts (where training data is more sparse).

## D  EVALUATION METRICS

**Accuracy.** The VQA dataset includes annotations from ten human reviewers per question. The accuracy of a given answer $a$ depends on how many of the provided answers it agrees with. It is scored as correct if at least 3 humans answers agree:

$$\text{Acc}(a) = \min\left[\frac{\#\text{humans that said } a}{3}, 1\right].\tag{20}$$

Each answer's accuracy is averaged over each 10-choose-9 set of human answers. As described in the main text, we only consider examples where the consensus answer was in the range of 0-20. We use all ten labels to calculate accuracy, regardless of whether individual labels deviate from this range. Thee accuracy values we report are taken from the average accuracy over some set of examples.

**RMSE.** This metric simply quantifies the typical deviation between the model count and the ground-truth. Across a set of $N$, we calculate this metric as

$$\text{RMSE} = \sqrt{\frac{1}{N}\sum_i (\hat{C}_i - C_i)^2},\tag{21}$$

where $\hat{C}_i$ and $C_i$ are the predicted and ground truth counts, respectively, for question $i$. RMSE is a measurement of error, so lower is better.

**Grounding Quality.** We introduce a new evaluation method for quantifying how relevant the objects counted by a model are to the type of object it was asked to count. This evaluation metric takes advantage of the ground truth labels included in the COCO dataset. These labels annotate each object instance of 80 different categories for each of the images in the development set. We make use of GloVe embeddings to compute semantic similarity. We use $\texttt{GloVe}(x) \in \mathbb{R}^{300}$ to denote the ($L2$ normalized) GloVe embedding of category $x$.

For each image $m$, the analysis is carried out in two stages.

First, we assign one of the COCO categories (or background) to each of the object proposals used for counting. For each object proposal, we find the object in the COCO labels with the largest IoU. If the IoU is above 0.5, we assign the object proposal to the category of the COCO object, otherwise we assign the object proposal to the background. Below, we use $k_i^m$ to denote the category assigned to object proposal $i$ for image $m$.

Second, for each of the COCO categories present in image $m$, we use the category $q$ (i.e. $q =$ "car") to build a question (i.e. "How many *cars* are there?"). For SoftCount and IRLC, the count returned in response to this question is the sum of each object proposal's inferred count value:

$$C^{(m,q)} = \sum_i^{N^m} w_i^{(m,q)},$$ (22)

where $N^m$ is the number of object proposals in image $m$ and $w_i^{(m,q)}$ is the count value given to proposal $i$. We use the count values to compute a weighted sum of the semantic similarity between the assigned object proposal categories $k$ and the question category $q$:

$$s^{(m,q)} = \sum_i^{N^m} w_i^{(m,q)} \left( \texttt{GloVe}\left(k_i^m\right)^T \texttt{GloVe}\left(q\right) \right),$$ (23)

where semantic similarity is estimated from the dot product between the embeddings of the assigned category and the question category. If $k_i^m$ corresponds to the background category, we replace its embedding with a vector of zeros.

The final metric is computed for each COCO category by accumulating the results over all images that contain a label for that category and normalizing by the net count to get an average:

$$s^{(q)} = \frac{\sum_m s^{(m,q)}}{\sum_m C^{(m,q)}}.$$ (24)

The interpretation of this metric is straightforward: on average, how relevant are the counted objects to the subject of the question.

