# OpenReview forum: "Interpretable Counting for Visual Question Answering"
_ICLR.cc/2018/Conference — Accept (Poster)_

### Official Review · AnonReviewer3 · 2017-11-26

**Rating:** 7
**Confidence:** 4

**Review:**


------------------
Summary:
------------------
This work introduces a discrete and interpretable model for answering visually grounded counting questions. The proposed model executes a sequential decision process in which it 1) selects an image region to "add to the count" and then 2) updates the likelihood of selecting other regions based on their relationships (defined broadly) to the selected region. After substantial module pre-trianing, the model is trained end-to-end with the REINFORCE policy gradient method (with the recently proposed self-critical sequence training baseline). Compared to existing approaches for counting (or VQA in general), this approach not only produces lower error but also provides a more human-intuitive discrete, instance-pointing representation of counting.

-----------------------
Preliminary Evaluation:
-----------------------
The paper presents an interesting approach that seems to outperform existing methods. More importantly in my view, the model treats counting as a discrete human-intuitive process. The presentation and experiments are okay overall but I have a few questions and requests below that I feel would strengthen the submission.

------------------
Strengths:
------------------
- I generally agree with the authors that approaching counting as a region-set selection problem provides an interpretable and human-intuitive methodology that seems more appropriate than attentional or monolithic approaches.

- To the best of my knowledge, the writing does a good job of placing the work in the context of existing literature.

- The dataset construction is given appropriate attention to restrict its instances to counting questions and will be made available to the public.

- The model outperforms existing approaches given the same visual and linguistic inputs / encodings. While I find improvements in RMSE a bit underwhelming, I'm still generally positive about the results given the improved accuracy and human-intuitiveness of the grounded outputs.

- I appreciated the analysis of the effect of "commonness" and think it provides interesting insight into the generalization of the proposed model.

- Qualitative examples are interesting.

------------------
Weaknesses:
------------------
- There is a lot going on in this paper as far as model construction and training procedures go. In its current state, many of the details are pushed to the supplement such that the main paper would be insufficient for replication. The authors also do not promise code release.

- Maybe it is just my unfamiliarity with it, but the caption grounding auxiliary-task feels insufficiently introduced in the main paper.  I also find it a bit discouraging that the details of joint training is regulated to the supplementary material, especially given that the UpDown is not using it! I would like to see an ablation of the proposed model without joint training.

- Both the IRLC and SoftCount models are trained with objectives that are aware of the ordinal nature of the output space (such that predicting 2 when the answer is 20 is worse than predicting 19). Unfortunately, the UpDown model is trained with cross-entropy and lacks access to this notion. I believe that this difference results in the large gap in RMSE between IRLC/SoftCount and UpDown. Ideally an updated version of UpDown trained under an order-aware loss would be presented during the rebuttal period. Barring that due to time constraints, I would otherwise like to see some analysis to explore this difference, maybe checking to see if UpDown is putting mass in smooth blobs around the predicted answer (though there may be better ways to see if UpDown has captured similar notions of output order as the other models).

- I would like to see a couple of simple baselines evaluated on HowMany-QA. Specifically, I think the paper would be stronger if results were put in context with a question only model and a model which just outputs the mean training count. Inter-human agreement would also be interesting to discuss (especially for high counts).

- The IRLC model has a significantly large (4x) capacity scoring function than the baseline methods. If this is restricted, do we see significant changes to the results?

- This is a relatively mild complaint. This model is more human-intuitive than existing approaches, but when it does make an error by selecting incorrect objects or terminating early, it is no more transparent about the cause of these errors than any other approach. As such, claims about interpretability should be made cautiously.

------------------
Curiosities:
------------------
- In my experience, Visual Genome annotations are often noisy, with many different labels being applied to the same object in different images. For per-image counts, I don't imagine this will be too troubling but was curious if you ran into any challenges.

- It looks like both IRLC and UpDown consistently either get the correct count (for small counts) or underestimate. This is not the Gaussian sort of regression error that we might expect from a counting problem.

- Could you speak to the sensitivity of the proposed model with respect to different loss weightings? I saw the values used in Section B of the supplement and they seem somewhat specific.

------------------
Minor errors:
------------------
[5.1 end of paragraph 2] 'that accuracy and RSME and not' -> 'that accuracy and RSME are not'
[Fig 9 caption] 'The initial scores are lack' -> 'The initial scores lack'

---

> ### Author Response · Authors · 2017-12-23
> **Revisions in response to reviewer 3**
>
> We would like to thank the reviewer for their thoughtful and constructive feedback. Before addressing the reviewer’s concerns, we should mention that since the original submission, we have observed superior performance (for all the models we consider) when using the visual features released with the original UpDown paper (those used by Anderson et al.). We believe this choice simplifies our work by focusing our contribution on the counting module and, by using publicly available visual features, facilitates future comparison.
>
> We have revised our paper to provide more detail about the influence of joint-training and have incorporated more detail into the main text. We also now compare each model with and without joint training and make it the default for each model to include joint training (since, with the new visual features, each model benefits from the extra supervision). These changes appear in Model section 4.2, Table 2 of the Results section, and Appendix Section B.1 in the revised submission.
>
> Unfortunately, we did not have sufficient time to update the training procedure of the UpDown model. Instead, we follow the reviewer’s suggestion and include an analysis of the smoothness of the model’s predictions. These changes appear in Appendix Section C of the revised submission.
>
> We have included the two simple baselines recommended by the reviewer. These changes appear in Results section 5.1 and Table 2 in the revised submission.
>
> Addressing the reviewer’s curiosities…
> The only issue noisy annotations in Visual Genome might cause would be with pre-training the vision module. Since we have opted to use publicly available pre-trained features, we ultimately don’t deal with that noise ourselves. However, the new features were pre-trained with Visual Genome and produce better results than our original attempt. We could speculate that our original features were worse because we did not account for the noisy labels, but that’s hard to say for certain.
> We have revised the paper to include the results of a small sweep over the loss weightings (Appendix Section C in revised submission).
>
> Anderson et al. Bottom-Up and Top-Down Attention for Image Captioning and VQA. In CVPR, 2017.

---

### Official Review · AnonReviewer2 · 2017-11-27
**Review from AnonReviewer2**

**Rating:** 7
**Confidence:** 4

**Review:**

This paper proposed a new approach for counting in VQA called Interpretable Counting in Visual Question Answering.  The authors create a new dataset (HowMany-QA) by processing the VQA 2.0 and Visual Genome dataset. In the paper, the authors use object detection framework (R-FCN) to extract bounding boxes information as well as visual features and propose three different strategies for counting. 1: SoftCount; 2: UpDown; 3: IRLC.  The authors show results on HowMany-QA dataset for the proposed methods, and the proposed IRLC method achieves the best performance among all the baselines.

[Strenghts]

This paper first introduced a cleaned visual counting dataset by processing existing VQA 2.0 and Visual Genome dataset, which can filter out partial non-counting questions. The proposed split is a good testbed for counting in VQA.

The authors proposed 3 different methods for counting, which both use object detection feature trained on visual genome dataset.  The object detector is trained with multiple objectives including object detection, relation detection, attribute classification and caption grounding to produce rich object representation. The author first proposed 2 baselines: SoftCount uses a Huber loss, UpDown uses a cross entropy loss. And further proposed interpretable RL counter which enumerates the object as a sequential decision process. The proposed IRLC more intuitive and outperform the previous VQA method  (UpDown) on both accuracy and RMSE.

[Weaknesses]

This paper proposed an interesting and intuitive counting model for VQA. However, there are several weaknesses existed:

1: The object detector is pre-trained with multiple objectives. However, there is no ablation study to show the differences. Since the model only uses the object and relationship feature as input, the authors could show results on counting with different  objects detector. For example, object detector trained using object + relation v.s. object + relation + attribute v.s. object + relation + attribute + caption.

2: Figure 9 shows an impressive result of the proposed method. Given the detection result, there are a lot of repetitive candidates detection bounding boxes. Without any strong supervision, IRLC could select the correct bounding boxes associated with the different objects. This is interesting, however, the authors didn't show any quantitative results on this. One experiment could verify the performance on IRLC is to compute the IOU between the GT COCO bounding box annotation on a small validation set. The validation set could be obtained by comparing the number of the bounding box and VQA answer with respect to similar COCO categories and VQA entities.

3: The proposed IRLC is not significantly outperform baseline method (SoftCount) with respect to RMSE (0.1). However, it would be interesting to see how the counting performance can change the result of object detection. As Chattopadhyay's CVPR2017 paper Sec 5.3 on the same subset as in point 2.

[Summary]

This paper proposed an interesting and interpretable model for counting in VQA. It formulated the counting as a sequential decision process that enumerated the subset of target objects. The authors introduce several new techniques in the IRLC counter. However, there is a lack of ablation study on the proposed model. Taking all these into account, I suggest accepting this paper if the authors could provide more ablation study on the proposed methods.

---

> ### Author Response · Authors · 2017-12-23
> **Revisions in response to reviewer 2**
>
> We would like to thank the reviewer for their thoughtful and constructive feedback. Before addressing the reviewer’s concerns, we should mention that since the original submission, we have observed superior performance (for all the models we consider) when using the visual features released with the original UpDown paper (those used by Anderson et al.). We believe this choice simplifies our work by focusing our contribution on the counting module and, by using publicly available visual features, facilitates future comparison.
>
> One consequence of this decision is that we forgo the option to do ablation studies on the vision module, as suggestion by the reviewer in comment 1. We very much agree that it will be interesting to see how these details of pre-training improve the model’s ability to map questions onto object representations. Therefore, in our revisions, we include a new ablation experiment on joint-training counting and caption grounding (changes appear in Table 2 of the Results section in the revised submission).
>
> In comment 2, the reviewer mentions the possibility of comparing the counting objects during question answering to the ground truth objects from the COCO labels. We thank the reviewer for this insightful suggestion and have included a new analysis to perform this comparison (changes appear in the Results section 5.2 and Appendix Section D in the revised submission). The results demonstrate that, despite the similarity between these two models with respect to RMSE, the objects counted by IRLC are more relevant to the question than the objects counted by SoftCount.
>
> The reviewer has also pointed out the possibility that our model could improve object detection. Because our model counts from questions, we believe our approach might be best suited to improve generalization of object detection to more diverse classes -- for example, to detect objects based not only on their class but also attributes and/or context in the scene. Time constraints prevent us from exploring this notion in the revised submission, but it is our goal to do so before a final submission.
>
> Anderson et al. Bottom-Up and Top-Down Attention for Image Captioning and VQA. In CVPR, 2017.
>
> Chattopadhyay et al. Counting Everyday Objects in Everyday Scenes. In CVPR, 2017.

---

> > ### Comment · AnonReviewer2 · 2018-01-16
> > **Post-rebuttal evaluation**
> >
> > I'm satisfied with the authors' responses to the concerns raised by me and my fellow reviewers, I would recommend acceptance of the paper.

---

### Official Review · AnonReviewer1 · 2017-11-28
**Size of test set too small, lacks comparison with existing models, paper writing needs to be improved.**

**Rating:** 6
**Confidence:** 3

**Review:**

Summary:
The paper presents a novel method for answering “How many …?” questions in the VQA datasets. Unlike previously proposed approaches, the proposed method uses an iterative sequential decision process for counting the relevant entity. The proposed model makes discrete choices about what to count at each time step. Another qualitative difference compared to existing approaches is that the proposed method returns bounding boxes for the counted object. The training and evaluation of the proposed model and baselines is done on a subset of the existing VQA dataset that consists of “How many …?” questions. The experimental results show that the proposed model outperforms the baselines discussed in the paper.

Strengths:
1.	The idea of sequential counting is novel and interesting.
2.	The analysis of model performance by grouping the questions as per frequency with which the counting object appeared in the training data is insightful.

Weaknesses:
1.	The proposed dataset consists of 17,714 QA pairs in the dev set, whereas only 5,000 QA pairs in the test set. Such a 3.5:1 split of dev and test seems unconventional. Also, the size of the test set seems pretty small given the diversity of the questions in the VQA dataset.
2.	The paper lacks quantitative comparison with existing models for counting such as with Chattopadhyay et al. This would require the authors to report the accuracies of existing models by training and evaluating on the same subset as that used for the proposed model. Absence of such a comparison makes it difficult to judge how well the proposed model is performing compared to existing models.
3.	The paper lacks analysis on how much of performance improvement is due to visual genome data augmentation and pre-training? When comparing with existing models (as suggested in above), this analysis should be done, so as to identify the improvements coming from the proposed model alone.
4.	The paper does not report the variation in model performance when changing the weights of the various terms involved in the loss function (equations 15 and 16).
5.	Regarding Chattopadhyay et al. the paper says that “However, their analysis was limited to the specific subset of examples where their approach was applicable.” It would be good it authors could elaborate on this a bit more.
6.	The relation prediction part of the vision module in the proposed model seems quite similar to the Relation Networks, but the paper does not mention Relation Networks. It would be good to cite the Relation Networks paper and state clearly if the motivation is drawn from Relation Networks.
7.	It is not clear what are the 6 common relationships that are being considered in equation 1. Could authors please specify these?
8.	In equation 1, if only 6 relationships are being considered, then why does f^R map to R^7 instead of R^6?
9.	In equations 4 and 5, it is not clarified what each symbol represents, making it difficult to understand.
10.	What is R in equation 15? Is it reward?

Overall:
The paper proposes a novel and interesting idea for solving counting questions in the Visual Question Answering tasks. However, the writing of the paper needs to be improved to make is easier to follow. The experimental set-up – the size of the test dataset seems too small. And lastly, the paper needs to add comparisons with existing models on the same datasets as used for the proposed model. So, the paper seems to be not ready for the publication yet.

---

> ### Author Response · Authors · 2017-12-23
> **Revisions in response to reviewer 1**
>
> We would like to thank the reviewer for their thoughtful and constructive feedback. Before addressing the reviewer’s concerns, we should mention that since the original submission, we have observed superior performance (for all the models we consider) when using the visual features released with the original UpDown paper (those used by Anderson et al.). We believe this choice simplifies our work by focusing our contribution on the counting module and, by using publicly available visual features, facilitates future comparison.
>
>
> Regarding comment 1: To examine the robustness of the test metrics, we re-computed the accuracy for the development and test splits after diverting 6500 randomly chosen QA pairs from dev to test (giving the adjusted dev/test splits 11k QA pairs each). We did this for the 8 IRLC models from the hyperparameter sweep whose penalty weights surrounded the optimum. On the original dev/test splits, those models have average accuracies of 56.21 & 57.06. In the adjusted split, the average accuracies are 56.18 & 56.64. This analysis suggests that the smaller test size does introduce some noise into the accuracy measurement, but the effect of that noise is small compared to the scale of the performance differences between SoftCount, UpDown and IRLC.
>
> Regarding comments 2, 3 and 5: The reviewer points out that we did not sufficiently place our work in the context of Chattopadhyay et al. We agree and have attempted to correct that mistake in our revised submission (changes appear in Related Works and Models [page 4] sections).
> To further clarify our reasoning here, there are three main reasons we do not compare to their work.
> 1) Our work and theirs both examine counting, but we use counting as a lens for exploring interpretability in visual question answering. This led to considerably different architectures as considered between our works.
> 2) Our work examines generalization for unseen or few-shot classes. Since the question is a sentence, not a single word, there are a number of cases where the effective class is a combination of noun and adjective or position (eg. black birds, people sitting on the bench). Chattopadhyay et al. only handles counting a fixed set of object classes and lacks the flexibility required for question answering.
> 3) The reviewer has suggested that we train and evaluate a model based on their proposals (i.e. “seq-sub”); however, to do so would make it very difficult to control for the quality/structure of visual features and the mechanism of visual-linguistic fusion. Additionally, the seq-sub architecture is not amenable to question answering or supervision from VQA data alone.
> All in all, we believe that the issues described above makes quantitatively comparing our work to that of Chattopadhyay et al. overly complicated and we hope the reviewer will agree with our assessment.
>
> As part of incorporating the new visual features, we have revised the model section describing the vision module. This revision has resulted in the removal of the confusing text that the reviewer mentioned in comments 6-8.
>
> Following this change, we cannot readily assess how details of pre-training affect ultimate performance, as recommended in comment 3. However, we have included an experiment to demonstrate the effect of data augmentation with Visual Genome (Appendix Section C in revised submission). We observe that removing the Visual Genome data reduces accuracy by 2.7% on average and increases RMSE by 0.12 on average and that IRLC is most robust to the decrease in training data.
>
> In addition, we have incorporated the experiment suggested in comment 4 (Appendix Section C in revised submission). The results demonstrate the range of weight settings in which the penalties improve performance.
>
> We have also clarified the model descriptions referred to in comments 9 and 10.
>
>
> Anderson et al. Bottom-Up and Top-Down Attention for Image Captioning and VQA. In CVPR, 2017.
>
> Chattopadhyay et al. Counting Everyday Objects in Everyday Scenes. In CVPR, 2017.

---

> > ### Comment · AnonReviewer1 · 2018-01-14
> > **Post-rebuttal evaluation**
> >
> > After reading the authors' responses to the concerns raised by me and my fellow reviewers, I would recommend acceptance of the paper because it presents a novel, interesting and interpretable method for counting.

---

### Public Comment · ~Sanyam_Agarwal1 · 2018-06-16
**ICLR 2018 Reproducibility Challenge report**

I could not find any publicly available implementation of this paper. So as a self-exercise I decided to take part in the ICLR 2018 Reproducibility Challenge and reproduce the results in this paper. Link to my implementation https://github.com/sanyam5/irlc-vqa

Summary of the report
==============================

Results (without caption grounding)

SoftCount
Model	                      Test Accuracy	       Test RMSE	   Training Time
Reported	                  49.2	                       2.45	              Unknown
This implementation	  49.7	                        2.31	             ~12 minutes (Nvidia-1080 Ti)

IRLC
Model	                       Test Accuracy 	Test RMSE	   Training Time
Reported	                       56.1                   	2.45              Unknown
This implementation	      55.7*	               2.41	             ~6 hours (Nvidia-1080 Ti)
*= Still improving. Work in Progress.

The accuracy was calculated using the VQA evaluation metric. I used the exact same script for calculating "soft score" as in https://github.com/hengyuan-hu/bottom-up-attention-vqa.

RMSE = root mean squared error from the ground truth (see below for how ground truth was chosen for VQA).

Note: These numbers correspond to the test accuracy and RMSE when the accuracy on the development set was maximum. The peak test accuracy is usually higher by about a percent.

# Key differences from the paper
GRU was used instead of LSTM for generating question embeddings. Experiments with LSTM led to slower learning and more over-fitting. More hyper-parameter search is required to fix this.

Gated Tanh Unit is not used. Instead, a 2-layer Leaky ReLu based network inspired by https://github.com/hengyuan-hu/bottom-up-attention-vqa with slight modifications is used.

# Filling in missing details in the paper
VQA Ground Truth
I couldn't find any annotations for a "single ground truth" which is requred to calculate the REINFORCE reward in IRLC. Also, I could not find any details in the paper relating to this issue. So I took as ground truth the label that was reported as the answer most number of times. In case there are more than one such label, the one having the least numerical value was picked (this might explain a lower RMSE).

Number of epochs
The authors mentioned that they use early stopping based on the development set accuracy but I couldn't find an exact method to determine when to stop. So I run the training for 100 epochs for IRLC and 20 epochs for SoftCount.

Number of candidate objects
I could not find the value of N = number of candidate objects that are taken from Faster-R-CNN so following https://github.com/hengyuan-hu/bottom-up-attention-vqa I took N=36.

# Minor discrepancies
Number of images due to Visual Genome
From Table 1 in the paper, it would seem that adding the extra data from Visual Genome doesn't change the number of training images (31932). However, while writing the dataloaders for Visual Genome I noticed around 45k images after including the visual genome dataset. This is not really a big issue, but I still thought I'd write it so that other people can avoid wasting their time investigating it.

---

### Decision · Program_Chairs · 2018-01-29
**ICLR 2018 Conference Acceptance Decision**

**Decision:**

Accept (Poster)

**Comment:**

Important problem and all reviewers recommend acceptance. I agree.